behaviour, evolution, ecology

imperfect mimicry, evolution, Batesian mimicry, ant mimic, locomotor mimicry, multicomponent mimicry

**Author for correspondence:**
Donald James McLean
e-mail: jim_mclean@optusnet.com.au

# Mimicry in motion and morphology: do information limitation, trade-offs or compensation relax selection for mimetic accuracy?

Donald James McLean and Marie E. Herberstein

Department of Biological Sciences, Macquarie University, Sydney, New South Wales 2109, Australia

DJM, 0000-0001-6229-7063; MEH, 0000-0001-5071-2952

Many animals mimic dangerous or undesirable prey as a defence from predators. We would expect predators to reliably avoid animals that closely resemble dangerous prey, yet imperfect mimics are common across a wide taxonomic range. There have been many hypotheses suggested to explain imperfect mimicry, but comparative tests across multiple mimicry systems are needed to determine which are applicable, and which—if any—represent general principles governing imperfect mimicry. We tested four hypotheses on Australian ant mimics and found support for only one of them: the information limitation hypothesis. A predator with incomplete information will be unable to discriminate some poor mimics from their models. We further present a simple model to show that predators are likely to operate with incomplete information because they forage and make decisions while they are learning, so might never learn to properly discriminate poor mimics from their models. We found no evidence that one accurate mimetic trait can compensate for, or constrain, another, or that rapid movement reduces selection pressure for good mimicry. We argue that information limitation may be a general principle behind imperfect mimicry of complex traits, while interactions between components of mimicry are unlikely to provide a general explanation for imperfect mimicry.

## 1. Introduction

Mimicry is the phenotypic resemblance of a mimic to a model. Mimicry comprises one or more signals that have been selected—by signal receivers—for their similarity to corresponding signals or cues in their models [1]. In this context, the signal conveys information that elicits a behavioural response in the receiver [2]. Traditional mimicry theory assumed that perfect mimicry was the optimal phenotype. Recent theory has recognized that mimetic resemblance need only be good enough to fool the signal receiver [3,4]. However, when signal receivers suffer the loss of fitness from failing to discriminate mimics from models (e.g. in deceptive mimicry), the optimal signal receiver behaviour is perfect discrimination [5]. Thus, the existence of imperfect mimics can be considered an evolutionary puzzle, since either the mimetic resemblance and/or receiver behaviour is apparently not optimal [6].

Researchers have proposed numerous explanations for imperfect mimicry. They can be broadly grouped, based (to some extent) on the adaptive landscapes they describe for mimics: human perception, evolving, constraints, trade-offs, relaxed selection, perceptual or cognitive exploitation and kin selection [6]. Several of these hypotheses have been extensively tested, while others are still largely untested for two main reasons: (1) they are relatively recent; and (2) they are difficult to test because they incorporate multiple signal components [6]. Our aim here is to focus on those lesser-tested hypotheses, the most recent of which is the information limitation hypothesis [7]. While most

**Table 1.** Hypotheses addressed by this study and whether our results support the hypothesis. The Exclusive column indicates whether the prediction is exclusive to the hypothesis, hence can be used to discriminate between this and the other tested hypotheses.

| hypothesis | prediction | exclusive? | ref | supported? |
|---|---|---|---|---|
| information limitation | more mimics are mistakenly classified as ants when the analysis uses incomplete information | yes | [7] | yes |
| motion-limited discrimination | poor mimics move faster or spend more time moving than good mimics | no | [8] | no |
| multicomponent hypotheses | | | | |
| increased deception | accuracy in two signals is negatively correlated | no | [2] | no |
| multitasking | accuracy in two signals is negatively correlated | no | [2] | no |
| backup signal[a] | accuracy in two signals is positively correlated | yes | [2] | no |
| receiver variability[a] | accuracy in two signals is not correlated | yes | [2] | yes |

[a]The backup signal and receiver variability hypotheses are general multicomponent signal hypotheses that do not address imperfect mimicry.

theoretical approaches to imperfect mimicry assume that receivers operate with the complete knowledge needed to make optimal decisions, the information limitation hypothesis considers that optimal receiver behaviour may be to operate with incomplete information, rather than to bear the cost of further sampling [7]. Receivers avoid potentially risky sampling of mimics and thus operate with insufficient information to accurately discriminate between poor mimics and their models. This increases the likelihood of mistaking mimics for their models, thereby relaxing selection on poor mimics [7]. Another recently proposed and related hypothesis argues that selection may be relaxed if mimics are in constant motion, thereby preventing receivers from adequately assessing visual similarity between mimic and model [8]. We label this the 'motion-limited discrimination' hypothesis.

Mimicry can consist of multiple signals, potentially across a range of sensory modalities [3]. Little is known about how multiple signals interact, although recent studies have generated multiple hypotheses addressing these interactions [2]. If one convincing mimetic signal (e.g. behavioural mimicry) compensates for another less convincing mimetic signal (e.g. morphological mimicry), the 'increased deception' hypothesis [2,9] argues that this could explain the imperfect mimicry of the second signal. Accordingly, the hypothesis predicts a negative association between the accuracy of two mimetic signals [10]. Similarly, if a mimic's ability to generate one signal is constrained by its generation of another (the 'multitasking' hypothesis [2]), this could also relax selection and result in imperfect mimicry of one signal, and again predicts a negative correlation between the two signals. A positive relationship between pairs of mimetic signals will result if selection for accuracy acts similarly on both signals (the 'backup signal' hypothesis [2,9]). No relationship is predicted if different signals are directed at different receivers (the 'receiver variability' hypothesis [2]). Importantly, neither the backup signal nor the receiver variability hypotheses are able to explain imperfect mimicry.

Mimicry has been described across a wide taxonomic range [11], with ant mimicry being one of the most commonly encountered forms [12]. A combination of characteristics such as aggressiveness, chemical defences, low nutritional value, group defence and conspicuousness makes ants ideal models for defensive (or Batesian) mimicry [13]. Consequently, ant mimicry has evolved independently many times, across as

many as 54 arthropod families [12]. Mimetic traits include body shape, colour and colour pattern, surface texture, size and behaviour [12]. Behaviour may be the most conspicuous feature of ants, and is, therefore, likely to be mimicked [14]. Ant mimics may behaviourally resemble their models in multiple ways, such as abdominal bobbing, emulating antennae by waving a pair of legs [15] and in the trajectories they follow while walking (locomotor mimicry) [16].

The aim of this study is to test several recent hypotheses explaining imperfect mimicry, using locomotor and morphological ant mimicry as a model system. We assess a number of poorly tested hypotheses, some mutually exclusive and some non-exclusive, that pertain to mimicry signal content and the maintenance of inaccurate mimicry (table 1). Using ant-mimicking spiders and insects and non-mimicking arthropods we quantify one behavioural trait (the trajectories followed while walking) and one morphological trait (body shape) and generate accuracy scores by comparing them to ants. While trajectory and body shape are each comprised a set of multiple traits, we calculate a single score for each set, allowing them to be compared as simple traits.

We then apply these accuracy scores to assess the prediction of the information limitation hypothesis: that receivers who incompletely sample potential prey items will fail to distinguish some imperfect mimics from models. Consequently, we expect significantly more mimics will be classified as ants by a limited information analysis than by a full information analysis. While not explicitly predicted by the information limitation hypothesis, we expect that misclassifications of ants and other non-mimetic arthropods will not change significantly. In addition, since the information limitation hypothesis assumes incomplete learning by receivers, we use a digital simulation of a simple learning model applied to our trajectory data to explore the conditions under which the hypothesis might apply. While logic dictates that greater statistical sampling will increase accuracy, our primary aim is to demonstrate this in a mimic/model context. Furthermore, this type of hypothesis testing facilitates the identification of real-world mimic encounter rates that can increase or decrease selection on the accuracy of mimicry.

While our data are unable to distinguish between the increased deception and multitasking hypotheses, we test their shared prediction: that the accuracy of two mimetic signals should be inversely correlated. We further assess the

predictions of the backup signal (a positive correlation between signals) and receiver variability (no relationship) hypotheses. Finally, we assess the motion-limited discrimination hypothesis by testing the prediction that poor mimics move faster or more consistently than good mimics.

## 2. Material and methods

### (a) Animals

A variety of ants and ant-mimicking spiders and insects were collected along the east coast of Australia, from Port Douglas, Queensland, to Sydney, New South Wales, between April 2017 and November 2018. We also collected arthropods that were not considered to be mimics in the literature but occurred in the same locations and habitats as—and had similar body sizes to—the ants and mimics (we refer to these arthropods as non-mimics). Specimens were collected from trees and bushes during daylight hours by visual inspection, searching under loose bark and by beating vegetation into sorting trays. Spiders and ants were housed individually in 50 ml plastic jars containing damp cotton wool for moisture, and spiders were fed twice per week on fruit flies. Lists of the specimens used for trajectory and morphometric analysis are available in CSV format in the electronic supplementary material. For information on the taxonomic identification, please see electronic supplementary material, appendix S2.

### (b) Trajectory analysis

We used the paths followed by walking animals—their trajectories—to assess behavioural mimicry. Here, we briefly summarize the steps followed to characterize trajectories in preparation for analysis; the process is fully described in electronic supplementary material, appendix S3. First, animals were filmed walking on a vertical featureless board in the laboratory. A vertical board was considered more ecologically relevant than a horizontal board as the majority of specimens were collected from tree trunks or vegetation. Next, custom software was used to extract trajectory coordinates from the video files and write them to CSV files. Finally, the extracted trajectories were numerically characterized using several indices that represented aspects of speed or movement and straightness or sinuosity (see electronic supplementary material, appendix S3 for details), with all trajectory quantification performed using the R package trajr [17,18]. The result is a vector of numbers that describes each trajectory and can be used for trajectory analysis. Sample sizes are summarized in electronic supplementary material, tables S1 and S2.

We used discriminant analysis to classify the characterized trajectories as either 'ant-like' or 'not ant-like'. This analysis identifies the information available to discriminate between the trajectories of ants and those of other animals. The output is the 'predicted' classification, derived from the properties of the trajectory. As the data were not homoscedastic (electronic supplementary material, appendix S5), we used quadratic discriminant analysis. All discriminant analyses were conducted with equal (i.e. uninformative) priors, and with cross validation to prevent overfitting and make the predictions conservative, unless otherwise noted.

Testing the motion-limited and multicomponent hypotheses (increased deception, multitasking, backup signal and receiver variability) required continuous-valued accuracy scores, with high values indicating highly ant-like trajectories, and low values indicating trajectories that were not ant-like. We applied logistic regression to the characterized trajectories to obtain these values, using logit-scaled predicted values as mimetic accuracy scores.

### (c) Morphological accuracy

Geometric morphometric analysis was used to characterize the dorsal and lateral outlines of animals. Outlines were obtained by photographing specimens dorsally and laterally, then manually tracing the outlines in Adobe Photoshop. To increase our sample size, we also used outlines from Kelly et al. [19]. Elliptical Fourier analysis was applied to the outlines, resulting in a numeric vector describing each shape (for further details, see [19]). The Fourier analysis produces output vectors with many dimensions, so we applied a principal components analysis both to remove constant dimensions and for dimension reduction, retaining 95% of the variation. Finally, using the same logic as for characterized trajectories, we applied discriminant analysis to classify shapes as ant- or non-ant-like, and logistic regression to calculate accuracy scores. Sample sizes are summarized in electronic supplementary material, tables S1 and S2.

### (d) Hypothesis testing

According to the information limitation hypothesis, some apparently imperfect mimics may be functionally perfect mimics when discrimination is based on limited information [7]. To test this, we compared the performance of an analysis that uses all available information against an analysis that uses limited information. To calculate accuracy based on full information, the discriminant analysis was trained on all samples (models, mimics and non-mimics). To calculate accuracy with incomplete information, the analysis was only trained on the model and non-mimic trajectories, representing a predator that only samples prey that are very likely to be encountered, and avoids potentially risky mimics. Cross validation was not used for incomplete analysis, since it is used to classify trajectories that were not part of the training set. We used a Pearson's $\chi^2$-test for independence to compare the proportion of mimics misclassified as ants by the fully trained discriminant analysis to the proportion misclassified by the partially trained analysis ($\alpha =$ 0.05). Pearson's $\chi^2$-test was also used to assess changes in misclassifications of ants and non-mimics.

To test for the relationship between the accuracy of multiple signals, we needed a numeric value for mimetic accuracy rather than a binary ant/not-ant decision, so we used logistic regression rather than discriminant analysis. We calculated the mean accuracy per species for both locomotor mimicry and morphological mimicry. We then fitted a linear regression to assess the type (positive or negative) and strength of the relationship between the two aspects of mimicry. We used linear least squares to test the relationship between mean and maximum walking speed while moving (i.e. excluding times when the animal was stopped) and morphological accuracy, both averaged to species ($\alpha = 0.05$). Due to the small sample size, we calculated bootstrap estimates of the 95% confidence intervals of the correlation coefficients.

### (e) Predator learning simulation

When Sherratt & Peet-Paré [7] proposed the information limitation hypothesis, they observed that predators must make decisions even while learning to discriminate between prey and models. To investigate the effect of learning on prey discrimination, we simulated a decision-making-while-learning process such as a naive predator may undergo. Every time a prey item is encountered, the predator's current knowledge of desirable and undesirable prey is used to decide whether to attack. If the prey is attacked, the predator's decision criteria are updated based on the actual type of prey. If it is ignored, the predator's criteria do not change. Until sufficient desirable and undesirable prey have been encountered, all prey are assumed to be desirable. We ignored other factors that would affect real predators' choices such as hunger, apparent prey availability and limited memory.

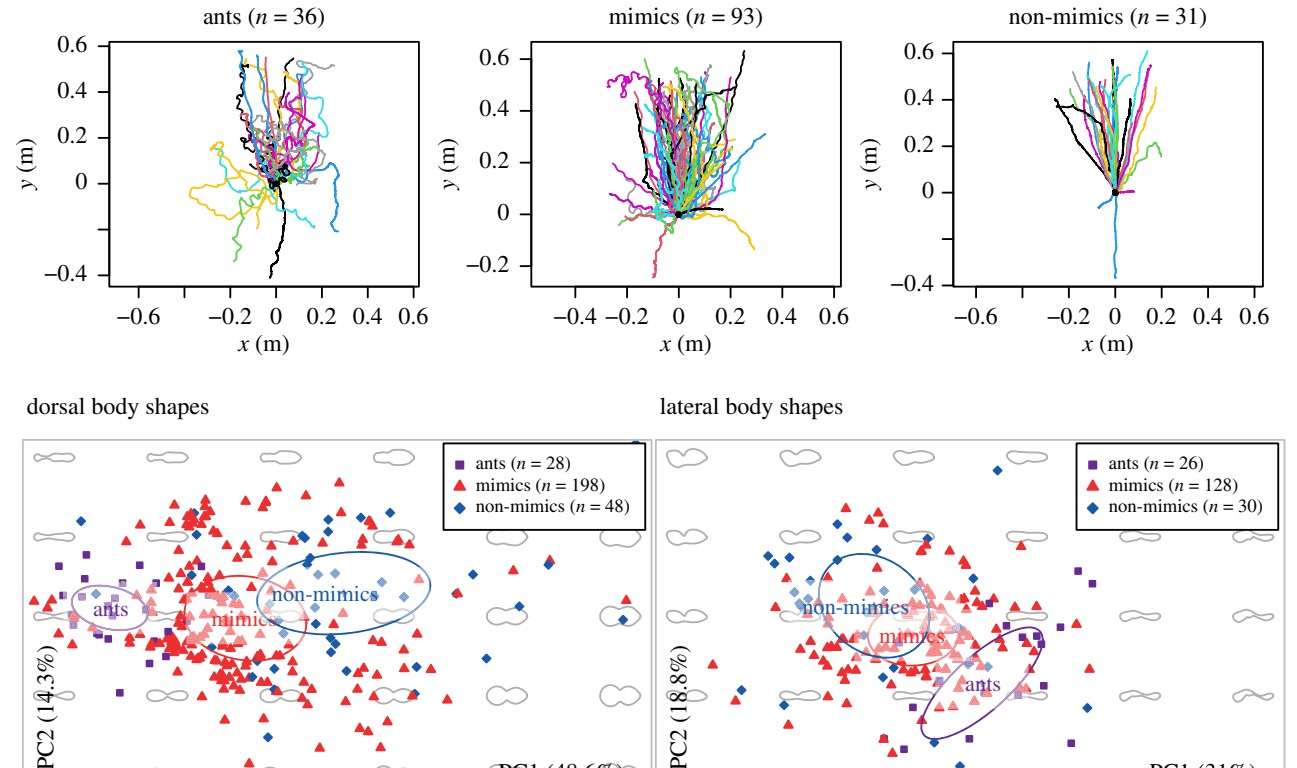

**Figure 1.** Trajectories (top row) and body shapes (bottom row) used in the analysis. Trajectory colours are randomly assigned. Dorsal and lateral body shapes are shown as points in two-dimensional morphospace. The grey background shapes indicate how shapes vary along the morphospace axes. (Online version in colour.)

We created a digital simulation of this model, presenting defined proportions of different prey types (mimics, models and non-mimics) to predators in random order. After each attack, we evaluated how the predator classified all potential prey. The simulated prey were trajectories, randomly selected (with replacement) from our samples, and the prey evaluation was performed as described above for trajectory analysis; trajectories classified as ants were avoided and those classified as non-ants were attacked. After each attack, we re-evaluated the mimetic accuracy of all trajectories using a discriminant analysis trained on just the trajectories that had been attacked so far, to determine the proportion of trajectories that would be considered desirable prey at that point in time. We repeated the simulation 1000 times and averaged the results to obtain the proportion of mimics, models and non-mimics that would be attacked after every encounter, then plotted the result. We performed this analysis under two hypothetical scenarios: mimics were abundant in the first scenario at 33% of total prey, and rare in the second at 5% of total prey. Models and non-mimics were equally abundant in both scenarios. The first scenario might apply to more relatively abundant mimics such as hoverflies [20], while the second could represent relative abundances of ant mimics (D.J.M. 2018, 2019, personal observation).

All tests and simulations were implemented in R [18].

## 3. Results

We characterized 160 trajectories from 58 species or morphospecies, comprised 93 mimic trajectories, 36 ant trajectories and 31 non-mimic trajectories (figure 1). Body lengths ranged from below 3 mm up to 16 mm, although mimetic spiders were all shorter than 7 mm. We characterized the body outlines of 304 specimens: 210 mimics, 42 ants and 52 non-mimics (figure 1). We could not always obtain both lateral and dorsal outlines for specimens, so our dataset contained 274 dorsal outlines and 184 lateral outlines

(figure 1). We obtained both trajectories and morphometric data for 15 species of ant mimics (electronic supplementary material, table S2).

### (a) Information limitation

When trained using all trajectories (i.e. using full information), 74% of mimics and 94% of non-mimics were correctly classified as suitable prey, while 47% of were ants incorrectly classified as potential prey (table 2). When trained on ants and non-mimics only (i.e. with limited information), identification of mimics as prey reduced to 56% (i.e. more mimics were avoided), incorrect identification of ants decreased to 3% and non-mimic identifications were unchanged at 94% (table 2). Mimics were significantly more likely to be misclassified as ants by the discriminant analysis trained on limited data (Pearson's $\chi^2 = 6.1$, d.f. = 1, $p = 0.01$). Misclassifications of ants were reduced significantly in the limited analysis (Pearson's $\chi^2 = 16.7$, d.f. = 1, $p < 0.001$). Misclassifications of non-mimics were unchanged.

Analysis of dorsal body shapes based on full information correctly identified 99% of mimics and 100% of non-mimics as suitable prey, and mistakenly identified 7% of ants as prey. With limited information, 85% of mimics were classified as prey, while all ants and non-mimics were correctly identified (table 2). As with trajectories, limited information resulted in mimics being significantly more likely to be misclassified as ants (Pearson's $\chi^2 = 23.7$, d.f. = 1, $p < 0.001$). Although the two ants incorrectly classified by the full information analysis were classified correctly by the limited information analysis, the change was not significant (Pearson's $\chi^2 = 0.5$, d.f. = 1, $p = 0.47$). All non-mimics were correctly identified by both analyses. Lateral body shape analysis also showed similar results. Correct classifications of mimics as prey were reduced significantly from 100% to

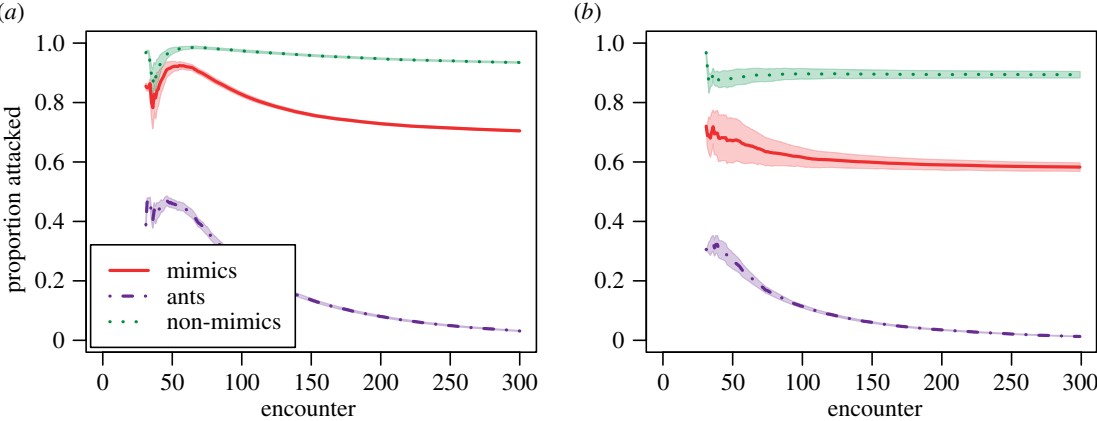

**Figure 2.** Proportion attacked for different prey types during simulated learning. Predators encounter equal numbers of ants and non-mimics, and encounter mimics at a rate of either (a) 33% or (b) 5% of total prey. Optimal predator behaviour is to attack all mimics (solid red line) and non-mimics (dashed green) and avoid all ants (dot-dash purple). Predators are initially naive and attack all prey (not shown). As they encounter ants, they start to build up criteria for prey items to be avoided. As each prey is encountered, predators either attack and update selection criteria, or ignore, in which case selection criteria remain unchanged. The decision to attack or not is based on information derived from previous attacks. After each encounter, the proportions of all prey items that would be attacked are recalculated. Means ± sample variance from 1000 simulations are shown. (Online version in colour.)

**Table 2.** Proportions of trajectories or body shapes identified as suitable prey (i.e. not ants), with full or limited information. Limited information consistently results in an increased rate of misclassifications of mimics, i.e. more mimics are mistaken for ants. However, limited information also generally improves the classification of ants. Orange backgrounds indicate that more mistakes were made with limited information, and blue backgrounds indicate fewer mistakes. Statistically significant changes are in bold.

| trait | trained on | ants | mimics | non-mimics |
|---|---|---|---|---|
| trajectory | full | 17/36 (47%) | 69/93 (74%) | 29/31 (94%) |
| | limited | **1/36 (3%)** | **52/93 (56%)** | 29/31 (94%) |
| dorsal outline | full | 2/28 (7%) | 196/198 (99%) | 48/48 (100%) |
| | limited | 0/28 (0%) | **169/198 (85%)** | 48/48 (100%) |
| lateral outline | full | 0/26 (0%) | 128/128 (100%) | 29/30 (97%) |
| | limited | 0/26 (0%) | **117/128 (91%)** | 30/30 (100%) |

91% (Pearson's $\chi^2 = 9.5$, d.f. = 1, $p = 0.002$), all ants were correctly identified by both analyses, and 1 non-mimic was mistakenly classified as an ant by the full analysis, but not by the limited analysis.

## (b) Learning and information limitation

Our learning simulations showed that the learning process, together with relative prey abundances, can affect how predators choose which prey to attack (figure 2). Under this learning model, if mimics are relatively common, predators quickly gain enough information to identify them as prey in over 70% of potential encounters (figure 2a). A lower relative abundance of mimics leads to smaller proportions of prey being attacked for all prey types, but particularly for mimics, with 58% being attacked (figure 2b).

## (c) Multicomponent hypotheses

We related trajectory mimetic accuracy to body outline accuracy for 15 species. There was no significant correlation between behavioural mimetic accuracy and dorsal morphological mimetic accuracy (adjusted $r^2 = -0.006$, $p = 0.36$, 95% confidence interval for $r = [-0.4, 0.7]$, $n = 15$). Similarly, there was no significant correlation between behavioural accuracy

and lateral morphological accuracy (adjusted $r^2 = -0.02$, $p = 0.42$, 95% confidence interval for $r = [-0.3, 0.6]$, $n = 15$).

## (d) Motion-limited discrimination

No measure of speed that we tested correlated significantly with morphometric accuracy. We tested mean speed (dorsal accuracy: adjusted $r^2 = -0.04$, $p = 0.5$, 95% confidence interval for $r$ (CI) = $[-0.5, 0.7]$; lateral accuracy: adjusted $r^2 = -0.07$, $p = 0.9$, 95% CI = $[-0.6, 0.5]$), maximum speed (dorsal accuracy: adjusted $r^2 = -0.04$, $p = 0.5$, CI = $[-0.7, 0.5]$; lateral accuracy: adjusted $r^2 = -0.04$, $p = 0.5$, CI = $[-0.7, 0.4]$), and proportion of time moving (dorsal accuracy: adjusted $r^2 = -0.07$, $p = 0.9$, CI = $[-0.4, 0.5]$; lateral accuracy: adjusted $r^2 = -0.07$, $p = 0.8$, CI = $[-0.6, 0.5]$). Our findings for each of the tested hypotheses is summarized in table 1.

## 4. Discussion

### (a) Information limitation

We used discriminant analysis to test the information limitation hypothesis, predicting that limited information about mimics would result in a higher rate of misclassifications of mimic trajectories and body shapes, and our results support

the prediction (table 2). Initially surprising, however, was the finding that limited information can also result in a significantly lower rate of misclassification of ant trajectories (reduced from 47% to 3%, table 2). This means that predators are more likely to avoid potentially costly attacks on ants when they operate with limited information than when they operate with full information. While our results support the information limitation hypothesis, it is worth noting that our analysis differs slightly from the original description of the hypothesis. The hypothesis was explained in terms of salient traits 'overshadowing' other, less salient traits [7], whereas we have shown that the interpretation of a single complex trait (such as trajectory or body shape) may depend on the amount of information available to the predator. Our results also indicate that while predators operating with limited information may reduce attacks on imperfect mimics, a large proportion of mimics can be distinguished from their models even with incomplete information. This means that information limitation does not explain all imperfect mimicry.

## (b) Learning and information limitation

The information limitation hypothesis was explicitly framed in terms of predator learning, and argued that exploration– exploitation models might describe predator behaviour [7]. However, exploration–exploitation models are computationally intractable for many real-world problems. Our learning simulation shows how a naive predator following one simple rule—avoid anything that seems to be an ant based on what is already known—unavoidably operates with incomplete information under some circumstances (figure 2). By following this rule, predators improve their decision criteria (i.e. they learn) when they misclassify an ant, but not when they misclassify, and hence avoid, mimics or nonmimics. If mimics are relatively uncommon within the prey community, predators will not obtain the information required to identify them, resulting in many being misclassified as models, thereby relaxing selection for more accurate mimicry. If, however, predators encounter many mimics, particularly during the early stages of learning, they operate with more complete information, and are able to correctly classify most mimics. An experiment with wild birds and artificial prey found exactly this relationship—imperfect mimics are better protected at low relative abundances than at high relative abundances [21]. The authors concluded that higher encounter rates are required to learn to distinguish imperfect mimics from their models, which informally describes the information limitation hypothesis.

In one sense our information limitation result is obvious: we have shown that a discriminant analysis with insufficient information performs poorly, and that poorer information leads to poorer performance. This unsurprising fact is not a quirk of the way that discriminant analysis behaves, but rather the nature of a complex discrimination task: any discrimination mechanism will perform poorly with insufficient information. In particular, predators or other signal receivers are subject to this limitation. The performance of our model is likely to be close to optimal within the simulated ecological constraints. In practice, predators will sample less before attempting to avoid ants, they will forget over time, and they may use cognitive short cuts such as relying on a subset of detectable traits. Hence, we consider that predators are likely to perform worse than our model,

mistakenly avoiding a higher proportion of mimics. The importance of adequate sampling to effective decision making has only been applied recently to the ecology of mimicry [7], and our learning model shows how this limitation could apply to predators in practice, leading to relaxed selection for accuracy in mimics.

## (c) Multicomponent hypotheses

We looked at the relationship between the accuracy of locomotor mimicry and morphological mimicry to test the predictions of some multicomponent mimicry hypotheses. If a good signal compensates for a poor signal (the increased deception hypothesis [2]), or if the generation of one signal constrains another (the multitasking hypothesis [2]), the two signals will be negatively correlated. Spider locomotion depends on muscles contained within the legs, but also on hydraulic power developed by muscles in the head, and thus hydraulic power depends on head morphology [22]. Indeed, the constricted morphology of ant-mimicking spiders limits their ability to jump [22], so it would not be unreasonable to expect similar constraints on locomotion, or the inverse: ant-like locomotion constraining the evolution of mimetic morphology. However, we found that accuracy of morphology and locomotion were not negatively correlated, failing to support either hypothesis. The two mimetic signals were also not positively correlated, which would have suggested that the same selective forces are acting on both signals (the backup signal hypothesis [2]). Instead, we found no relationship, suggesting that the signals are directed at different receivers (the receiver variability hypothesis [2]). Of course, we only compared two signal types, so we cannot rule out correlations between other pairs of signals, however, this lack of correlation between behavioural and morphological mimicry is largely consistent with previous results for hoverflies [9].

## (d) Motion-limited discrimination

Finally, we investigated whether mimics that move at high speeds, or rarely stop, prevent accurate discrimination by predators, so faster or more active mimics would be less accurate morphological mimics—the motion-limited discrimination hypothesis [8]. Our data did not support this hypothesis, as we found no correlation between speed (or proportion of time moving) and morphological mimicry. Previous work found that poor morphological ant mimics had higher escape speeds than more accurate mimics. The higher speeds increased the likelihood of escape, thus reducing selection for greater mimetic accuracy [10]. Broadly similar results have been found using human 'predators', where the ability to escape reduced the likelihood of mimicry evolving [23]. Our findings may differ from these because we measured the speed of walking while undisturbed, rather than escape speed.

## 5. Conclusion

Do our results help reveal general principles of imperfect mimicry? Information limitation is not specific to ant mimics but will apply to any complex mimetic signal that cannot be learnt by sampling a small number of instances. While locomotor mimicry might be a particularly 'complex' trait in this sense, it seems likely that many other forms of

mimicry are also complex, hence the information limitation hypothesis is likely to be widely applicable. On the other hand, the multicomponent hypotheses tested here depend on the specific characteristics of the mimic, model or predator, with limited capacity to describe general principles.

Imperfect mimicry seems to be a sub-optimal phenotype, but current thinking is that since mimics only need to be good enough to fool signal receivers, imperfect mimicry is expected [3,4]. This thinking only moves the problem: rather than asking why the mimic phenotype is not optimal, we now need to ask why signal receiver discrimination or behaviour is not optimal. The information limitation hypothesis provides a possible answer to this question: an evolved improvement in cognitive power or visual acuity cannot overcome the lack of information brought about by ecological circumstances. Some imperfect mimicry may simply result from unavoidable ecological constraints.

**Data accessibility.** Data (including trajectory videos and outline images) and code for this study are available from the Dryad Digital Repository: https://doi.org/10.5061/dryad.15dv41nwj [24] and GitHub (https://github.com/JimMcL/mimicry-in-motion). Appendices (in PDF format) and specimen and species lists (in CSV format) are provided as electronic supplementary material.

**Authors' contributions.** D.J.M.: Conceptualization, data curation, formal analysis, investigation, methodology, software, writing-original draft; M.E.H.: conceptualization, funding acquisition, project administration, supervision, validation, writing-review and editing

Both authors gave final approval for publication and agreed to be held accountable for the work performed therein.

**Competing interests.** We declare we have no competing interests.

**Funding.** This work was supported by the Australian Research Council [Project ID: DP170101617].

**Acknowledgements.** Many thanks to Zoe Wild for collecting, filming, and caring for specimens, Sue Downes for collecting specimens, Michael Kelly and Dylan Geraghty for filming specimens, Lizzy Lowe for reviewing drafts and Dr Drew Allen for statistics advice. Thanks also to the two anonymous reviewers and the editor for their helpful comments.

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
