## [Peer Review File · Proceedings of the Royal Society B: Biological Sciences]

Review History

RSPB-2021-0135.R0 (Original submission)

Review form: Reviewer 1

Recommendation

Major revision is needed (please make suggestions in comments)

Scientific importance: Is the manuscript an original and important contribution to its field?

Acceptable

General interest: Is the paper of sufficient general interest?

Good

Quality of the paper: Is the overall quality of the paper suitable?

Marginal

Is the length of the paper justified?

Yes

Should the paper be seen by a specialist statistical reviewer?

No

Do you have any concerns about statistical analyses in this paper? If so, please specify them explicitly in your report.

Yes

It is a condition of publication that authors make their supporting data, code and materials available - either as supplementary material or hosted in an external repository. Please rate, if applicable, the supporting data on the following criteria.

Is it accessible?

N/A

Is it clear?

N/A

Is it adequate?

N/A

Do you have any ethical concerns with this paper?

No

Comments to the Author

This manuscript explores several hypotheses that may account for imperfect mimicry in an ant mimicry system. The approaches are novel and there are some interesting ideas considered in the discussion. However, while the material in this manuscript is interesting, it is very densely written and significant content is lost in the difficulty of understanding what is being read. Thus, my main concern is that, as written, the material is inaccessible to readers.

Here, a number of simulation-based approaches are used to determine how an animal might respond to an ant mimic versus and ant or another type of insect (although I can find nowhere where any information about this is given, either in the manuscript or in the appendix). This is a novel approach, but currently exceedingly hard to follow. Even when reading the discussion, I am having to go back literally every sentence to the results to see if I can find the correlation between what is stated in the discussion and the results, and I find that I often cannot. In my view, this is a rare example where I think the manuscript would significantly benefit from having results and discussion together, such that the results are stated and interpreted in the same block, allowing for a better understanding of often apparently counterintuitive (if I am reading these correctly) results.

In the abstract it is somewhat misleading to suggest a broad comparative approach and then discuss an ant mimetic system. This is a great model system, but perhaps be a bit cautious with the wording in the abstract. I think as it is stated in the discussion, this is fine, but perhaps the abstract should pertain more specifically to what you do show.

L. 36. I think this sentence should be paraphrased to be a bit more nuanced. Newer models, including some which you cite, (e.g., Dalzeill et al. 2016, *Ecol Letters*; de Jager & Anderson 2019, *Functional Ecology*), suggest that perfect resemblance is not the optimal phenotype, but instead enough of a match of a percept in the receiver to be 'fooled'. With some careful re-writing, I think you can still make the arguments that you make in this manuscript, while acknowledging these newer frameworks - indeed, I think your incomplete information would be bolstered by this.

L. 76 "Using ant-mimicking spiders and insects and non-mimicking arthropods" is a bit vague... doesn't make a whole lot of sense.

L1, Page 7 (on pdf; line numbering has disappeared). Remove "a" before "indices"

I can find no information on the species investigated in the main article or in the appendices. Seems like this information is much needed. Indeed, I found some of the methods a bit hard to follow with respect to sample sizes and species (e.g., how many of each species and how many species were used to calculate trajectories? - for ants, for mimics, and for whatever the other group consisted of).

I am having a bit of trouble following the text in the results. You say “Analysis of dorsal body shapes based on full information correctly identified 99% of mimics, 93% of ants and 100% of non-mimics, changing to 94%, 100% and 100% respectively with limited information...limited information resulted in mimics being significantly more likely to be misclassified as ants”. My lack of understanding stems from say the change from 99 to 94% correct identification of the mimics under full vs limited information. Are the mimics being “correctly” identified as mimics? (as the text appears to suggest), or as ants? Both here and in the trajectory section, I think really clear writing to make explicit to the reader which direction is which would be really handy (this is a recurring issue throughout the manuscript).

I am a bit confused by Fig. 1, which seems to indicate that mimics are attacked less often when they are frequently encountered than when they are infrequently encountered. You state “A lower relative abundance of mimics leads to smaller proportions of prey being attacked for all prey types, but particularly for mimics” - I assume by prey here you mean mimics and non-ants (this should be made clear in the manuscript at the outset), so what you are saying is that with low frequency of mimics, palatable prey are attacked less often, and with high mimic frequency of mimics, palatable prey are attacked more often, and there is no effect on unpalatable prey (i.e., ants) in either case. I don't understand this at all...or am I just completely not getting the point? Either way, this probably means that the writing needs to be made clearer...

Discussion, line 2. Add comma before ‘predicting’

“Our results further suggest that, depending on the relative costs and benefits of attacking ants and mimics, and their relative abundances, predators may also make selectively advantageous prey choice decisions when operating with limited information, as they are more likely to avoid ants.” This is not apparent from Figure 1, and I can't see text in the results that states this. This sort of issue would be solved with a results and discussion merger, as suggested above.

Review form: Reviewer 2

Recommendation

Major revision is needed (please make suggestions in comments)

Scientific importance: Is the manuscript an original and important contribution to its field?

Excellent

General interest: Is the paper of sufficient general interest?

Good

Quality of the paper: Is the overall quality of the paper suitable?

Excellent

Is the length of the paper justified?

Yes

Should the paper be seen by a specialist statistical reviewer?

No

Do you have any concerns about statistical analyses in this paper? If so, please specify them explicitly in your report.

No

It is a condition of publication that authors make their supporting data, code and materials available - either as supplementary material or hosted in an external repository. Please rate, if applicable, the supporting data on the following criteria.

Is it accessible?

Yes

Is it clear?

Yes

Is it adequate?

Yes

Do you have any ethical concerns with this paper?

No

Comments to the Author

While some mimetic species bare a shocking morphological and/or behavioral resemblance to their models, other mimics seem to show a less-than-perfect level of similarity. These “imperfect mimics” have long puzzled biologists because they seem to represent a non-optimal state in a system that one might expect to be under steep selective pressure – with predators seeking to accurately differentiate profitable (edible) prey from non-profitable targets. While the question of imperfect mimicry has been the source of much theoretical work, there are surprisingly few cases where hypotheses have been tested directly.

This manuscript sets out to do precisely this. First, the authors provide a succinct overview of imperfect mimicry and the related leading hypotheses. Then, working with Australian ants, ant-mimics (especially spiders), and non-mimetic arthropods, they quantify morphology (dorsal and lateral body shape) and behavioral traits (a suite of characteristics related to overall locomotion) and evaluate numerous hypotheses in light of this data. They present three central findings: (1) using a predator learning simulation, their work suggests that imperfect mimicry may be a consequence of predators operating with limited information; (2) based on measured traits, it does not appear that accurate traits compensate for inaccurate traits, results that directly address proposed hypotheses; and (3) it does not appear that mimic speed during normal locomotion is correlated with scores of morphological mimicry, again directly addressing standing hypotheses.

The rigorous and straightforward hypothesis-testing approach that the authors have taken is commendable – and is particularly so when working on mimicry-related questions. I was especially impressed by the clarity with which the overarching theoretical frameworks and specific hypotheses were presented. The discussions of the relationship between tests and hypotheses, as well as the authors interpretations of the results and overall conclusions were all excellent, as well – including being appropriately nuanced, when necessary. Thus, in terms of overall conception, organization, topic, and quality, I believe that the manuscript is a good fit for Proceedings B.

However, I felt that a few things were missing from the manuscript (and the associated supplemental) that make evaluating the work difficult. While some are relatively minor, others (particularly item # 1) are not. I have outlined these, below.

Again, I think that the work and the manuscript are quite good, and I hope that the authors are willing to address the concerns that I have raised.

(1) Crucially, there is only minimal information available regarding the species that were used in the study. General counts of species categories are provided in the first section of the results, but details are not provided. Without this, it is impossible for the reader to understand the context of results and discussion. Personally, I believe that I can not fully evaluate the manuscript without this information.

If the authors could provide a table (or some other form of summary) showing the taxonomic descriptions and the data collected for each sample, that would be tremendously helpful. I understand that there are a large number of samples (300+), and this might need to be part of the supplemental, but without this information the reader is not able to evaluate (and appreciate) many of the results. For example, one of the central findings of the manuscript is that in 15 mimetic species where morphological and behavioral data could be collected, there was no significant correlation between these two feature sets. This result is interpreted in light of multi-component theory, suggesting a lack of support for two potential hypotheses about imperfect mimicry (the increased deception hypothesis and the multitasking hypothesis). However, because it is not clear what species are used, it is not clear how the reader should think about this result—for example, in the broad context of understanding imperfect mimicry, if the mimics are all from the same genus it suggests a rather different scope than if they are not.

(2) I also believe that the work would benefit from an additional figure showing some examples of the traits that were measured (for example, sample trajectories and body outlines). Perhaps also a PCA of shape outlines highlighting ants, mimics, and non-mimics in a 2D shape-space? Such additions would give readers who are new to ant-mimicry a better appreciation for the phenomena, and would give other mimicry researchers a more direct view onto the observed variation and the data used throughout the paper.

Minor comments:

(3) For the trajectory quantification, I am curious why a vertical platform was used. Horizontal surfaces have been used in other works and it would seem that horizontal surfaces are dominant in most naturally occurring contexts, with some notable exceptions such as tree-climbing. Was this done because of the ecology of the animals or for experimental reasons? A portion of the supplement addressing this, and/or an additional sentence in the main text would be helpful.

(4) In the Results concerning motion limited discrimination, the only relationships between speed and shapes are addressed. It seems that some measure of percent time moving might also be important here, as predators might be limited in a more thresholded way, where predator perception is blurred in a more categorical fashion (clearly visible when stationary, not clearly visible when moving) as opposed to on a continuous scale only when the target is in motion.

(5) My understanding is that the trajectory analysis portion of this work was done using *trajr*. I would suggest replacing the sentence:

“Finally, the extracted trajectories were numerically characterised using a several indicies that represented aspects of speed or movement and straightness or sinuosity [16].”

With something like: “Finally, the extracted trajectories were numerically characterised using several indicies that represent aspects of speed or movement and straightness or sinuosity (see supplemental for details), with all trajectory quantification performed using the *r*-package *trajr* [16].”

This makes it a more clear how the work was done, gives people a direct path to *trajr*, and also sends the interested reader to the supplement for full information about the various parameters that were quantified and the sources in the literature for those parameters.

(6) There seems to be an error in Figure 1. The text suggests that Fig. 1A shows results for when the model is relatively common (mimic encounter rate = 30%), corresponding to a relatively high

level of predation, and that Fig. 1B shows results for when the mimic is much less common (mimic encounter rate = 5%) which corresponds to a lower predation rate. However, the caption in Fig. 1 reads "A) 5% or B) 30%". Can the authors please verify which is correct?

Decision letter (RSPB-2021-0135.R0)

26-Feb-2021

Dear Mr McLean:

I am writing to inform you that your manuscript RSPB-2021-0135 entitled "Mimicry in motion and morphology: do information limitation, trade-offs or compensation relax selection for mimetic accuracy?" has, in its current form, been rejected for publication in Proceedings B.

This action has been taken on the advice of referees, who have recommended that substantial revisions are necessary. With this in mind we would be happy to consider a resubmission, provided the comments of the referees are fully addressed. However please note that this is not a provisional acceptance.

Sincerely,
Dr Sasha Dall
mailto: proceedingsb@royalsociety.org

Associate Editor
Comments to Author:

This study tests a subset of hypotheses for the evolution of imperfect mimicry – a long-standing puzzle of broad interest. There are many strengths of the study, including the clarity of the theoretical framework and hypotheses in the introduction, the rigorous hypothesis-testing approach, the novelty and sophistication of the approach, the fascinating study system, and the breadth of taxonomic sampling (and therefore ability to generalize). However, I and both expert

reviewers found it difficult, if not impossible, to properly evaluate the manuscript with the information provided. Reasons for this include the difficulty of relating the interpretation/discussion to the results presented (reviewer 1) and lack of information on the sampling (species used and number of trajectories or specimens for morphological analysis per species). Without this information, it is not possible to assess potential biases in the data. This is important because it affects the statistical sampling of the data and therefore potentially the discriminant analysis/main results.

In addition to comments by the two reviewers, I have a few other concerns:

1. In terms of the significance of the study, the knowledge gaps need to be articulated more clearly in the introduction. The study focuses on a couple of hypotheses but mentions that there are numerous hypotheses. It was not clear to me whether the hypotheses tested are the most widely accepted (or obscure), and whether they have been tested in other taxa. If there are many hypotheses to explain imperfect mimicry, are these the most prevalent? Why is it important to assess these particular hypotheses and have they been tested in other taxa?

2. Both the test of the information limitation hypothesis and learning simulation are based on sampling of data for discriminant analyses (sampling for the training dataset and comparison of classification when trained on different datasets). The assumption is that statistical sampling and discrimination is a good proxy for the information available to a predator and how the information is used (i.e. how a predator would sample and discriminate prey). Whether this assumption is likely to hold really needs to be discussed.

3. Related to 2 above, my greatest concern is that the results may follow necessarily from the statistical approach used to test mimetic accuracy. A DA finds the linear combination of variables that best distinguish pre-defined categories. Statistical performance will increase with sample size then plateau, and this will depend on sampling of each of the categories. This is what we see in figure 1. A similar argument can be made for the information limitation – better sampling means better statistical performance (i.e. discrimination). If the test of the information limitation hypothesis is to compare discrimination based on statistical sampling then it will necessarily show that discrimination improves with better sampling. And if the results are a forgone conclusion, then the approach used can only confirm rather than test the hypotheses. Essentially, I need to be convinced that the logic of the approach isn't circular.

Overall, both reviewers felt that the study has potential to be suitable for Proc B, and I agree, but the issues identified would need to be addressed before it can be properly evaluated.

Reviewer(s)' Comments to Author:

Referee: 1

Comments to the Author(s)

This manuscript explores several hypotheses that may account for imperfect mimicry in an ant mimicry system. The approaches are novel and there are some interesting ideas considered in the discussion. However, while the material in this manuscript is interesting, it is very densely written and significant content is lost in the difficulty of understanding what is being read. Thus, my main concern is that, as written, the material is inaccessible to readers.

Here, a number of simulation-based approaches are used to determine how an animal might respond to an ant mimic versus and ant or another type of insect (although I can find nowhere where any information about this is given, either in the manuscript or in the appendix). This is a novel approach, but currently exceedingly hard to follow. Even when reading the discussion, I am having to go back literally every sentence to the results to see if I can find the correlation between what is stated in the discussion and the results, and I find that I often cannot. In my view, this is a rare example where I think the manuscript would significantly benefit from having results and discussion together, such that the results are stated and interpreted in the same block,

allowing for a better understanding of often apparently counterintuitive (if I am reading these correctly) results.

In the abstract it is somewhat misleading to suggest a broad comparative approach and then discuss an ant mimetic system. This is a great model system, but perhaps be a bit cautious with the wording in the abstract. I think as it is stated in the discussion, this is fine, but perhaps the abstract should pertain more specifically to what you do show.

L. 36. I think this sentence should be paraphrased to be a bit more nuanced. Newer models, including some which you cite, (e.g., Dalzeill et al. 2016, *Ecol Letters*; de Jager & Anderson 2019, *Functional Ecology*), suggest that perfect resemblance is not the optimal phenotype, but instead enough of a match of a percept in the receiver to be 'fooled'. With some careful re-writing, I think you can still make the arguments that you make in this manuscript, while acknowledging these newer frameworks - indeed, I think your incomplete information would be bolstered by this.

L. 76 "Using ant-mimicking spiders and insects and non-mimicking arthropods" is a bit vague...doesn't make a whole lot of sense.

L1, Page 7 (on pdf; line numbering has disappeared). Remove "a" before "indices"

I can find no information on the species investigated in the main article or in the appendices. Seems like this information is much needed. Indeed, I found some of the methods a bit hard to follow with respect to sample sizes and species (e.g., how many of each species and how many species were used to calculate trajectories? - for ants, for mimics, and for whatever the other group consisted of).

I am having a bit of trouble following the text in the results. You say "Analysis of dorsal body shapes based on full information correctly identified 99% of mimics, 93% of ants and 100% of non-mimics, changing to 94%, 100% and 100% respectively with limited information...limited information resulted in mimics being significantly more likely to be misclassified as ants". My lack of understanding stems from say the change from 99 to 94% correct identification of the mimics under full vs limited information. Are the mimics being "correctly" identified as mimics? (as the text appears to suggest), or as ants? Both here and in the trajectory section, I think really clear writing to make explicit to the reader which direction is which would be really handy (this is a recurring issue throughout the manuscript).

I am a bit confused by Fig. 1, which seems to indicate that mimics are attacked less often when they are frequently encountered than when they are infrequently encountered. You state "A lower relative abundance of mimics leads to smaller proportions of prey being attacked for all prey types, but particularly for mimics" - I assume by prey here you mean mimics and non-ants (this should be made clear in the manuscript at the outset), so what you are saying is that with low frequency of mimics, palatable prey are attacked less often, and with high mimic frequency of mimics, palatable prey are attacked more often, and there is no effect on unpalatable prey (i.e., ants) in either case. I don't understand this at all...or am I just completely not getting the point? Either way, this probably means that the writing needs to be made clearer...

Discussion, line 2. Add comma before 'predicting'

"Our results further suggest that, depending on the relative costs and benefits of attacking ants and mimics, and their relative abundances, predators may also make selectively advantageous prey choice decisions when operating with limited information, as they are more likely to avoid ants." This is not apparent from Figure 1, and I can't see text in the results that states this. This sort of issue would be solved with a results and discussion merger, as suggested above.

Referee: 2

Comments to the Author(s)

While some mimetic species bare a shocking morphological and/or behavioral resemblance to their models, other mimics seem to show a less-than-perfect level of similarity. These “imperfect mimics” have long puzzled biologists because they seem to represent a non-optimal state in a system that one might expect to be under steep selective pressure – with predators seeking to accurately differentiate profitable (edible) prey from non-profitable targets. While the question of imperfect mimicry has been the source of much theoretical work, there are surprisingly few cases where hypotheses have been tested directly.

This manuscript sets out to do precisely this. First, the authors provide a succinct overview of imperfect mimicry and the related leading hypotheses. Then, working with Australian ants, ant-mimics (especially spiders), and non-mimetic arthropods, they quantify morphology (dorsal and lateral body shape) and behavioral traits (a suite of characteristics related to overall locomotion) and evaluate numerous hypotheses in light of this data. They present three central findings: (1) using a predator learning simulation, their work suggests that imperfect mimicry may be a consequence of predators operating with limited information; (2) based on measured traits, it does not appear that accurate traits compensate for inaccurate traits, results that directly address proposed hypotheses; and (3) it does not appear that mimic speed during normal locomotion is correlated with scores of morphological mimicry, again directly addressing standing hypotheses.

The rigorous and straightforward hypothesis-testing approach that the authors have taken is commendable – and is particularly so when working on mimicry-related questions. I was especially impressed by the clarity with which the overarching theoretical frameworks and specific hypotheses were presented. The discussions of the relationship between tests and hypotheses, as well as the authors interpretations of the results and overall conclusions were all excellent, as well – including being appropriately nuanced, when necessary. Thus, in terms of overall conception, organization, topic, and quality, I believe that the manuscript is a good fit for Proceedings B.

However, I felt that a few things were missing from the manuscript (and the associated supplemental) that make evaluating the work difficult. While some are relatively minor, others (particularly item # 1) are not. I have outlined these, below.

Again, I think that the work and the manuscript are quite good, and I hope that the authors are willing to address the concerns that I have raised.

(1) Crucially, there is only minimal information available regarding the species that were used in the study. General counts of species categories are provided in the first section of the results, but details are not provided. Without this, it is impossible for the reader to understand the context of results and discussion. Personally, I believe that I can not fully evaluate the manuscript without this information.

If the authors could provide a table (or some other form of summary) showing the taxonomic descriptions and the data collected for each sample, that would be tremendously helpful. I understand that there are a large number of samples (300+), and this might need to be part of the supplemental, but without this information the reader is not able to evaluate (and appreciate) many of the results. For example, one of the central findings of the manuscript is that in 15 mimetic species where morphological and behavioral data could be collected, there was no significant correlation between these two feature sets. This result is interpreted in light of multi-component theory, suggesting a lack of support for two potential hypotheses about imperfect mimicry (the increased deception hypothesis and the multitasking hypothesis). However, because it is not clear what species are used, it is not clear how the reader should think about this result – for example, in the broad context of understanding imperfect mimicry, if the mimics are all from the same genus it suggests a rather different scope than if they are not.

(2) I also believe that the work would benefit from an additional figure showing some examples of the traits that were measured (for example, sample trajectories and body outlines). Perhaps also a PCA of shape outlines highlighting ants, mimics, and non-mimics in a 2D shape-space? Such additions would give readers who are new to ant-mimicry a better appreciation for the phenomena, and would give other mimicry researchers a more direct view onto the observed variation and the data used throughout the paper.

Minor comments:

(3) For the trajectory quantification, I am curious why a vertical platform was used. Horizontal surfaces have been used in other works and it would seem that horizontal surfaces are dominant in most naturally occurring contexts, with some notable exceptions such as tree-climbing. Was this done because of the ecology of the animals or for experimental reasons? A portion of the supplement addressing this, and/or an additional sentence in the main text would be helpful.

(4) In the Results concerning motion limited discrimination, the only relationships between speed and shapes are addressed. It seems that some measure of percent time moving might also be important here, as predators might be limited in a more thresholded way, where predator perception is blurred in a more categorical fashion (clearly visible when stationary, not clearly visible when moving) as opposed to on a continuous scale only when the target is in motion.

(5) My understanding is that the trajectory analysis portion of this work was done using trajr. I would suggest replacing the sentence:

“Finally, the extracted trajectories were numerically characterised using a several indicies that represented aspects of speed or movement and straightness or sinuosity [16].”

With something like: “Finally, the extracted trajectories were numerically characterised using several indicies that represent aspects of speed or movement and straightness or sinuosity (see supplemental for details), with all trajectory quantification performed using the r-package trajr [16].”

This makes it a more clear how the work was done, gives people a direct path to trajr, and also sends the interested reader to the supplement for full information about the various parameters that were quantified and the sources in the literature for those parameters.

(6) There seems to be an error in Figure 1. The text suggests that Fig. 1A shows results for when the model is relatively common (mimic encounter rate = 30%), corresponding to a relatively high level of predation, and that Fig. 1B shows results for when the mimic is much less common (mimic encounter rate = 5%) which corresponds to a lower predation rate. However, the caption in Fig. 1 reads “A) 5% or B) 30%”. Can the authors please verify which is correct?

Author's Response to Decision Letter for (RSPB-2021-0135.R0)

See Appendix A.

RSPB-2021-0815.R0

Review form: Reviewer 1

Recommendation

Accept with minor revision (please list in comments)

Scientific importance: Is the manuscript an original and important contribution to its field?
Excellent

General interest: Is the paper of sufficient general interest?
Excellent

Quality of the paper: Is the overall quality of the paper suitable?
Good

Is the length of the paper justified?
Yes

Should the paper be seen by a specialist statistical reviewer?
No

Do you have any concerns about statistical analyses in this paper? If so, please specify them explicitly in your report.
No

It is a condition of publication that authors make their supporting data, code and materials available - either as supplementary material or hosted in an external repository. Please rate, if applicable, the supporting data on the following criteria.

Is it accessible?
Yes

Is it clear?
Yes

Is it adequate?
Yes

Do you have any ethical concerns with this paper?
No

Comments to the Author

This manuscript is significantly clearer now. The authors are to be commended for dramatically improving the readability of this interesting piece of work. The results and discussion (having sections), in particular, are dramatically improved. Well done!

On that note, I have only very few minor comments, below:

Note placement of ref 5 on L 47 and 6 on L 54

L 107, add commas after 'Queensland'

L 197/200. You refer to Fig 1 supp material - this does not exist. I think you are referring simply to Fig. 1

Discussion (no line numbers): "Our learning simulation shows how a naïve predator following one simple rule—avoid anything that seems to be an ant based on what is already known—unavoidably operate" add "s" to "operate"

Review form: Reviewer 2

Recommendation

Accept with minor revision (please list in comments)

Scientific importance: Is the manuscript an original and important contribution to its field?

Good

General interest: Is the paper of sufficient general interest?

Good

Quality of the paper: Is the overall quality of the paper suitable?

Good

Is the length of the paper justified?

Yes

Should the paper be seen by a specialist statistical reviewer?

No

Do you have any concerns about statistical analyses in this paper? If so, please specify them explicitly in your report.

No

It is a condition of publication that authors make their supporting data, code and materials available - either as supplementary material or hosted in an external repository. Please rate, if applicable, the supporting data on the following criteria.

Is it accessible?

Yes

Is it clear?

Yes

Is it adequate?

Yes

Do you have any ethical concerns with this paper?

No

Comments to the Author

I found this revised version of the manuscript much improved and appreciate the authors' efforts. The additions and corrections that were made have addressed my main concerns – in particular, the addition of Figure 1, which gives the reader a better sense of the locomotor and morphological features being discussed, and the clarification of which species were used in the analysis. I do have a few corrections, suggestions, and comments regarding this new draft, but I believe that the manuscript (in a lightly revised form) would be worthy of publication in Proceedings B.

Specific comments/suggestions:

- Line 30 (abstract): The final portion of the abstract states: “while interactions between components of mimicry may limit mimetic accuracy in some mimicry systems.” Perhaps I have simply missed it (or am misinterpreting), but it is not clear to me what this statement is based on – specifically regarding the use of the term “limit”. The section subtitled “Multicomponent hypothesis” in the Discussion seems to clearly state that there is no correlation between traits –

and that the current study does not find support for the limitation-related hypothesis that there exists some constraint(s) between ant-like movements and ant-like morphologies. Could another phrase be used in the abstract to avoid this confusion?

- Line 47: It appears that the citation “[5]” shows up here by mistake. Looking at the document showing the changes that were implemented, it seems like this citation might have just accidentally escaped deletion when removing an older piece of text.
- Line 54: Likely the same problem as on Line 47, this time regarding “[6]” at the beginning of the sentence.
- Line 84-85: The behavioral and morphological traits are discussed here as though they are actually singular traits. It might help if it is made clear that they are sets of traits, and for each set a single score was calculated.
- Table 2: The separations between types of hypotheses in the text is excellent (information limitation, multicomponent hypotheses, and motion-limited hypotheses), but it would be great if that organization could also be present in this table. Without this separation, the table is a bit confusing, particularly because it appears that multiple hypotheses are mutually exclusive but are also supported by the results of the work—a contradiction that disappears when it is clear that these hypotheses address different levels of analysis.
- Discussion: “Motion limited discrimination” section title: Some places in the text the phrase “motion limited” is without a hyphen, while in other cases it is hyphenated (as in the 3rd line of the “Motion limited discrimination” section and in Table 2).
- Discussion, “Multicomponent hypotheses”, 4th-6th lines: I like the point being made here, but I think it might help to be a bit more explicit about where the relationship between locomotor power, muscles, and morphology. Something like: “Spider locomotion depends on muscles contained within the legs, but also hydraulic power developed by muscles in the head, and thus hydraulic power depends on head morphology.”
- Discussion, “Motion limited discrimination”, 4th line: I would recommend changing “consistency of motion” to “fraction of time spent moving,” or something similar. To me, consistency of motion suggests the pattern and/or frequency of stops, as opposed to the overall duration of movement.

Decision letter (RSPB-2021-0815.R0)

07-May-2021

Dear Mr McLean

I am pleased to inform you that your manuscript RSPB-2021-0815 entitled "Mimicry in motion and morphology: do information limitation, trade-offs or compensation relax selection for mimetic accuracy?" has been accepted for publication in Proceedings B.

The referee(s) have recommended publication, but also suggest some minor revisions to your manuscript. Therefore, I invite you to respond to the referee(s)' comments and revise your manuscript. Because the schedule for publication is very tight, it is a condition of publication that you submit the revised version of your manuscript within 7 days. If you do not think you will be able to meet this date please let us know.

NB. From April 1 2013, peer reviewed articles based on research funded wholly or partly by RCUK must include, if applicable, a statement on how the underlying research materials – such

as data, samples or models – can be accessed. This statement should be included in the data accessibility section.

[http://datadryad.org/submit?journalID=RSPB&manu=\(Document not available\)](http://datadryad.org/submit?journalID=RSPB&manu=(Document%20not%20available)) which will take you to your unique entry in the Dryad repository. If you have already submitted your data to dryad you can make any necessary revisions to your dataset by following the above link. Please see <https://royalsociety.org/journals/ethics-policies/data-sharing-mining/> for more details.

Sincerely,

Dr Sasha Dall

Associate Editor

Board Member

Comments to Author:

Thank-you for carefully revising the manuscript. I agree with both reviewers that the clarity and readability of the manuscript is improved. In particular, I appreciated the stronger rationale for the hypotheses tested and the approach used in the introduction.

In terms of sampling design, it was useful to include the raw datasets as supplementary material but these comprise several large and complex spreadsheets, which don't help the reader understand the sampling. The results section starts with a high-level summary of the sampling [160 trajectories from 58 species or morphospecies, comprised of 93 mimic 197 trajectories, 36 ant trajectories and 31 non-mimic trajectories; body outlines of 304 specimens: 210 mimics, 42 ants and 52 non-mimics; trajectories and morphometric data for 15 species of ant-mimics]. However, as highlighted in the previous review, the finer-grained sampling is important because it can produce biases in the bootstrapping. What is needed is a brief summary describing the sampling in the Methods. For example, for the trajectories, how many of the 58 species were mimics, ants and non-mimics (number of species in each category)? What was the mean, SD and range of trajectories per species in each of these categories? Please provide this detail (samples per species and species per category) for both trajectories and morphology, and the samples per species for the 15 species with both types of data. This information just requires a line or two so would not add much to total length.

Apart from adding this information, the reviewers have provided a number of suggestions to improve clarity. These suggestions are minor and straight forward to address.

I think this paper presents significant empirical and conceptual advances and will be an important paper in the field. I look forward to seeing it published.

Reviewer(s)' Comments to Author:

Referee: 1

Comments to the Author(s).

This manuscript is significantly clearer now. The authors are to be commended for dramatically improving the readability of this interesting piece of work. The results and discussion (having sections), in particular, are dramatically improved. Well done!

On that note, I have only very few minor comments, below:

Note placement of ref 5 on L 47 and 6 on L 54

L 107, add commas after 'Queensland'

L 197/200. You refer to Fig 1 supp material – this does not exist. I think you are referring simply to Fig. 1

Discussion (no line numbers): "Our learning simulation shows how a naïve predator following one simple rule— avoid anything that seems to be an ant based on what is already known— unavoidably operate" add "s" to "operate"

Referee: 2

Comments to the Author(s).

I found this revised version of the manuscript much improved and appreciate the authors' efforts. The additions and corrections that were made have addressed my main concerns—in particular, the addition of Figure 1, which gives the reader a better sense of the locomotor and morphological features being discussed, and the clarification of which species were used in the analysis. I do have a few corrections, suggestions, and comments regarding this new draft, but I believe that the manuscript (in a lightly revised form) would be worthy of publication in Proceedings B.

Specific comments/suggestions:

- Line 30 (abstract): The final portion of the abstract states: "while interactions between components of mimicry may limit mimetic accuracy in some mimicry systems." Perhaps I have simply missed it (or am misinterpreting), but it is not clear to me what this statement is based on—specifically regarding the use of the term "limit". The section subtitled "Multicomponent hypothesis" in the Discussion seems to clearly state that there is no correlation between traits—and that the current study does not find support for the limitation-related hypothesis that there exists some constraint(s) between ant-like movements and ant-like morphologies. Could another phrase be used in the abstract to avoid this confusion?
- Line 47: It appears that the citation "[5]" shows up here by mistake. Looking at the document showing the changes that were implemented, it seems like this citation might have just accidentally escaped deletion when removing an older piece of text.
- Line 54: Likely the same problem as on Line 47, this time regarding "[6]" at the beginning of the sentence.
- Line 84-85: The behavioral and morphological traits are discussed here as though they are actually singular traits. It might help if it is made clear that they are sets of traits, and for each set a single score was calculated.
- Table 2: The separations between types of hypotheses in the text is excellent (information limitation, multicomponent hypotheses, and motion-limited hypotheses), but it would be great if that organization could also be present in this table. Without this separation, the table is a bit confusing, particularly because it appears that multiple hypotheses are mutually exclusive but are also supported by the results of the work—a contradiction that disappears when it is clear that these hypotheses address different levels of analysis.
- Discussion: "Motion limited discrimination" section title: Some places in the text the phrase "motion limited" is without a hyphen, while in other cases it is hyphenated (as in the 3rd line of the "Motion limited discrimination" section and in Table 2).

- Discussion, “Multicomponent hypotheses”, 4th-6th lines: I like the point being made here, but I think it might help to be a bit more explicit about where the relationship between locomotor power, muscles, and morphology. Something like: “Spider locomotion depends on muscles contained within the legs, but also hydraulic power developed by muscles in the head, and thus hydraulic power depends on head morphology.”
- Discussion, “Motion limited discrimination”, 4th line: I would recommend changing “consistency of motion” to “fraction of time spent moving,” or something similar. To me, consistency of motion suggests the pattern and/or frequency of stops, as opposed to the overall duration of movement.

Author's Response to Decision Letter for (RSPB-2021-0815.R0)

See Appendix B.

Decision letter (RSPB-2021-0815.R1)

13-May-2021

Dear Mr McLean

I am pleased to inform you that your manuscript entitled "Mimicry in motion and morphology: do information limitation, trade-offs or compensation relax selection for mimetic accuracy?" has been accepted for publication in Proceedings B.

Your article has been estimated as being 8 pages long. Our Production Office will be able to confirm the exact length at proof stage.

Data Accessibility section

Open Access

Paper charges

Sincerely,
Proceedings B
mailto: proceedingsb@royalsociety.org

Appendix A

Response to referees

Manuscript RSPB-2021-0135: "Mimicry in motion and morphology: do information limitation, trade-offs or compensation relax selection for mimetic accuracy?"

3 Mar 2021

Associate Editor

Comments to Author:

This study tests a subset of hypotheses for the evolution of imperfect mimicry – a long-standing puzzle of broad interest. There are many strengths of the study, including the clarity of the theoretical framework and hypotheses in the introduction, the rigorous hypothesis-testing approach, the novelty and sophistication of the approach, the fascinating study system, and the breadth of taxonomic sampling (and therefore ability to generalize). However, I and both expert reviewers found it difficult, if not impossible, to properly evaluate the manuscript with the information provided. Reasons for this include the difficulty of relating the interpretation/discussion to the results presented (reviewer 1) and lack of information on the sampling (species used and number of trajectories or specimens for morphological analysis per species). Without this information, it is not possible to assess potential biases in the data. This is important because it affects the statistical sampling of the data and therefore potentially the discriminant analysis/main results.

Response: Thank you for your helpful comments and suggestions. We have addressed these concerns by rewording the discussion (and correcting a mistake) which should make the relation of the results and discussion more easily understood. We have also included full lists of specimens as supplementary material, and added a new figure (Figure 1) as suggested by reviewer 2 showing trajectories and a PCA plot of the morphometric results.

In addition to comments by the two reviewers, I have a few other concerns:

1. In terms of the significance of the study, the knowledge gaps need to be articulated more clearly in the introduction. The study focuses on a couple of hypotheses but mentions that there are numerous hypotheses. It was not clear to me whether the hypotheses tested are the most widely accepted (or obscure), and whether they have been tested in other taxa. If there are many hypotheses to explain imperfect mimicry, are these the most prevalent? Why is it important to assess these particular hypotheses and have they been tested in other taxa?

Response: Thank you for the helpful comments. We have now sharpened the introduction to articulate the knowledge gap more clearly and provide greater justification for the selected hypotheses.

2. Both the test of the information limitation hypothesis and learning simulation are based on sampling of data for discriminant analyses (sampling for the training dataset and comparison of classification when trained on different datasets). The assumption is that statistical sampling and discrimination is a good proxy for the information available to a predator and how the information is used (i.e. how a predator would sample and discriminate prey). Whether this assumption is likely to hold really needs to be discussed.

Response: Thank you for pointing this out; we agree and have added comments to the discussion.

3. Related to 2 above, my greatest concern is that the results may follow necessarily from the statistical approach used to test mimetic accuracy. A DA finds the linear combination of variables that best distinguish pre-defined categories. Statistical performance will increase with sample size then plateau, and this will depend on sampling of each of the categories. This is what we see in figure 1. A similar argument can be made for the information limitation – better sampling means better statistical performance (i.e. discrimination). If the test of the information limitation hypothesis is to compare discrimination based on statistical sampling then it will necessarily show that discrimination improves with better sampling. And if the results are a forgone conclusion, then the approach used can only confirm rather than test the hypotheses. Essentially, I need to be convinced that the logic of the approach isn't circular.

Response: We do, in part, agree with this argument. Our response is that 1) we still believe that, even if the outcome is logical, the idea still needs demonstration, and we are the first to do so here; 2) we present a learning model to demonstrate the likely ecological relevance of this statistical phenomenon; and 3) an information limitation approach has power to predict under what natural history, i.e. encounter rates of mimics to models, we should expect more or less perfect mimics. We now make this explicit in our rationale and discussion.

Overall, both reviewers felt that the study has potential to be suitable for Proc B, and I agree, but the issues identified would need to be addressed before it can be properly evaluated.

Reviewer(s)' Comments to Author:

Referee: 1

Comments to the Author(s)

This manuscript explores several hypotheses that may account for imperfect mimicry in an ant mimicry system. The approaches are novel and there are some interesting ideas considered in the discussion. However, while the material in this manuscript is interesting, it is very densely written and significant content is lost in the difficulty of understanding what is being read. Thus, my main concern is that, as written, the material is inaccessible to readers.

Here, a number of simulation-based approaches are used to determine how an animal might respond to an ant mimic versus and ant or another type of insect (although I can find nowhere where any information about this is given, either in the manuscript or in the appendix). This is a novel approach, but currently exceedingly hard to follow. Even when reading the discussion, I am having to go back literally every sentence to the results to see if I can find the correlation between what is stated in the discussion and the results, and I find that I often cannot. In my view, this is a rare example where I think the manuscript would significantly benefit from having results and discussion together, such that the results are stated and interpreted in the same block, allowing for a better understanding of often apparently counterintuitive (if I am reading these correctly) results.

Response: We understand the reviewer's concern and have now drawn a stronger link in the discussion between the results and our interpretation thereof, and added subheadings to the discussion. We have corrected an error in the caption to Figure 2 (which was Figure 1 in the original manuscript) that contributed to the confusion. We also include specimen lists in the supplementary material.

In the abstract it is somewhat misleading to suggest a broad comparative approach and then discuss an ant mimetic system. This is a great model system, but perhaps be a bit cautious with the wording in the abstract. I think as it is stated in the discussion, this is fine, but perhaps the abstract should pertain more specifically to what you do show.

Response: We have changed the abstract wording to no longer claim that general principles can be obtained from our results.

L. 36. I think this sentence should be paraphrased to be a bit more nuanced. Newer models, including some which you cite, (e.g., Dalzeill et al. 2016, Ecol Letters; de Jager & Anderson 2019, Functional Ecology), suggest that perfect resemblance is not the optimal phenotype, but instead enough of a match of a percept in the receiver to be 'fooled'. With some careful re-writing, I think you can still make the arguments that you make in this manuscript, while acknowledging these newer frameworks - indeed, I think your incomplete information would be bolstered by this.

Response: Thank you for raising this point. We have reworded the paragraph to take these newer frameworks into account. We also address this idea in the conclusion.

L. 76 "Using ant-mimicking spiders and insects and non-mimicking arthropods" is a bit vague...doesn't make a whole lot of sense.

Response: We have provided a more detailed explanation of what we mean by "non-mimics", and the supplementary information now includes lists of specimens used for trajectory and body shape analysis.

L1, Page 7 (on pdf; line numbering has disappeared). Remove "a" before "indices"

Response: Done.

I can find no information on the species investigated in the main article or in the appendices. Seems like this information is much needed. Indeed, I found some of the methods a bit hard to follow with respect to sample sizes and species (e.g., how many of each species and how many species were used to calculate trajectories? - for ants, for mimics, and for whatever the other group consisted of).

Response: We now report sample sizes in the new figure 1 as well as in specimen lists in the supplementary material. After making these changes to reporting, we also decided to tighten our data by removing some poor quality or repeated trajectories for individuals, and being a little more conservative with our classification of mimic. This change has not qualitatively changed our results. All the original data remain available as raw data in the archived repository.

I am having a bit of trouble following the text in the results. You say "Analysis of dorsal body shapes based on full information correctly identified 99% of mimics, 93% of ants and 100% of non-mimics, changing to 94%, 100% and 100% respectively with limited information....limited information resulted in mimics being significantly more likely to be misclassified as ants". My lack of understanding stems from say the change from 99 to 94% correct identification of the mimics under full vs limited information. Are the mimics being "correctly" identified as mimics? (as the text appears to suggest), or as ants? Both here and in the trajectory section, I think really clear writing to

make explicit to the reader which direction is which would be really handy (this is a recurring issue throughout the manuscript).

Response: We have rewritten the results to make them clearer. We now express results as proportion of animals classified as potential prey, which we feel will remove the ambiguity caused by reporting proportion correctly classified.

I am a bit confused by Fig. 1, which seems to indicate that mimics are attacked less often when they are frequently encountered than when they are infrequently encountered. You state “A lower relative abundance of mimics leads to smaller proportions of prey being attacked for all prey types, but particularly for mimics” – I assume by prey here you mean mimics and non-ants (this should be made clear in the manuscript at the outset), so what you are saying is that with low frequency of mimics, palatable prey are attacked less often, and with high mimic frequency of mimics, palatable prey are attacked more often, and there is no effect on unpalatable prey (i.e., ants) in either case. I don’t understand this at all....or am I just completely not getting the point? Either way, this probably means that the writing needs to be made clearer...

Response: We regret that the label of Figure 1 (now figure 2 in the revised manuscript) was incorrect. We hope that with this correction and the re-worded results, the manuscript is now easier to understand.

Discussion, line 2. Add comma before ‘predicting’

Response: Done.

“Our results further suggest that, depending on the relative costs and benefits of attacking ants and mimics, and their relative abundances, predators may also make selectively advantageous prey choice decisions when operating with limited information, as they are more likely to avoid ants.” This is not apparent from Figure 1, and I can’t see text in the results that states this. This sort of issue would be solved with a results and discussion merger, as suggested above.

Response: We have removed this sentence, as it reiterated a prior point. More generally, we now reference the relevant result table or figure from the discussion to better tie results and discussion together.

Referee: 2

Comments to the Author(s)

While some mimetic species bare a shocking morphological and/or behavioral resemblance to their models, other mimics seem to show a less-than-perfect level of similarity. These “imperfect mimics” have long puzzled biologists because they seem to represent a non-optimal state in a system that one might expect to be under steep selective pressure—with predators seeking to accurately differentiate profitable (edible) prey from non-profitable targets. While the question of imperfect mimicry has been the source of much theoretical work, there are surprisingly few cases where hypotheses have been tested directly.

This manuscript sets out to do precisely this. First, the authors provide a succinct overview of imperfect mimicry and the related leading hypotheses. Then, working with Australian ants, ant-mimics (especially spiders), and non-mimetic arthropods, they quantify morphology (dorsal and

lateral body shape) and behavioral traits (a suite of characteristics related to overall locomotion) and evaluate numerous hypotheses in light of this data. They present three central findings: (1) using a predator learning simulation, their work suggests that imperfect mimicry may be a consequence of predators operating with limited information; (2) based on measured traits, it does not appear that accurate traits compensate for inaccurate traits, results that directly address proposed hypotheses; and (3) it does not appear that mimic speed during normal locomotion is correlated with scores of morphological mimicry, again directly addressing standing hypotheses.

The rigorous and straightforward hypothesis-testing approach that the authors have taken is commendable—and is particularly so when working on mimicry-related questions. I was especially impressed by the clarity with which the overarching theoretical frameworks and specific hypotheses were presented. The discussions of the relationship between tests and hypotheses, as well as the authors' interpretations of the results and overall conclusions were all excellent, as well—including being appropriately nuanced, when necessary. Thus, in terms of overall conception, organization, topic, and quality, I believe that the manuscript is a good fit for Proceedings B.

However, I felt that a few things were missing from the manuscript (and the associated supplemental) that make evaluating the work difficult. While some are relatively minor, others (particularly item # 1) are not. I have outlined these, below.

Again, I think that the work and the manuscript are quite good, and I hope that the authors are willing to address the concerns that I have raised.

(1) Crucially, there is only minimal information available regarding the species that were used in the study. General counts of species categories are provided in the first section of the results, but details are not provided. Without this, it is impossible for the reader to understand the context of results and discussion. Personally, I believe that I can not fully evaluate the manuscript without this information.

If the authors could provide a table (or some other form of summary) showing the taxonomic descriptions and the data collected for each sample, that would be tremendously helpful. I understand that there are a large number of samples (300+), and this might need to be part of the supplemental, but without this information the reader is not able to evaluate (and appreciate) many of the results. For example, one of the central findings of the manuscript is that in 15 mimetic species where morphological and behavioral data could be collected, there was no significant correlation between these two feature sets. This result is interpreted in light of multi-component theory, suggesting a lack of support for two potential hypotheses about imperfect mimicry (the increased deception hypothesis and the multitasking hypothesis). However, because it is not clear what species are used, it is not clear how the reader should think about this result—for example, in the broad context of understanding imperfect mimicry, if the mimics are all from the same genus it suggests a rather different scope than if they are not.

Response: We now provide supplementary CSV files listing the specimens used in the trajectory and body shape analysis, and the species used for the multi-component hypothesis tests. Columns include calculated accuracy figures (using full information) and sample sizes. After generating these reports, we decided to remove some poor quality or duplicated trajectories from the analysis and be slightly more conservative in our classification of mimics. This has not qualitatively altered our results. All trajectories, including those newly rejected, are available in the online archive.

(2) I also believe that the work would benefit from an additional figure showing some examples of the traits that were measured (for example, sample trajectories and body outlines). Perhaps also a PCA of shape outlines highlighting ants, mimics, and non-mimics in a 2D shape-space? Such additions would give readers who are new to ant-mimicry a better appreciation for the phenomena, and would give other mimicry researchers a more direct view onto the observed variation and the data used throughout the paper.

Response: We have added a new Figure 1 that shows body shapes in 2D shape-space and trajectories grouped into ants, mimics and non-mimics.

Minor comments:

(3) For the trajectory quantification, I am curious why a vertical platform was used. Horizontal surfaces have been used in other works and it would seem that horizontal surfaces are dominant in most naturally occurring contexts, with some notable exceptions such as tree-climbing. Was this done because of the ecology of the animals or for experimental reasons? A portion of the supplement addressing this, and/or an additional sentence in the main text would be helpful.

Response: A vertical board was used as the majority of specimens were collected from tree trunks or vegetation, so a vertical board was considered more ecologically relevant than a horizontal board. We have added an explanatory sentence to the main text.

(4) In the Results concerning motion limited discrimination, the only relationships between speed and shapes are addressed. It seems that some measure of percent time moving might also be important here, as predators might be limited in a more thresholded way, where predator perception is blurred in a more categorical fashion (clearly visible when stationary, not clearly visible when moving) as opposed to on a continuous scale only when the target is in motion.

Response: We agree that this is a good idea, which we have implemented and added to the text (there was no correlation).

(5) My understanding is that the trajectory analysis portion of this work was done using trajr. I would suggest replacing the sentence:

“Finally, the extracted trajectories were numerically characterised using a several indicies that represented aspects of speed or movement and straightness or sinuosity [16].”

With something like: “Finally, the extracted trajectories were numerically characterised using several indicies that represent aspects of speed or movement and straightness or sinuosity (see supplemental for details), with all trajectory quantification performed using the r-package trajr [16].”

This makes it a more clear how the work was done, gives people a direct path to trajr, and also sends the interested reader to the supplement for full information about the various parameters that were quantified and the sources in the literature for those parameters.

Response: We agree and have updated the text accordingly.

(6) There seems to be an error in Figure 1. The text suggests that Fig. 1A shows results for when the model is relatively common (mimic encounter rate = 30%), corresponding to a relatively high level of predation, and that Fig. 1B shows results for when the mimic is much less common (mimic encounter rate = 5%) which corresponds to a lower predation rate. However, the caption in Fig. 1 reads “A) 5% or B) 30%”. Can the authors please verify which is correct?

Response: Thank you for bringing this to our attention. The caption was wrong and has now been corrected.

Appendix B

Comments to Author:

Thank-you for carefully revising the manuscript. I agree with both reviewers that the clarity and readability of the manuscript is improved. In particular, I appreciated the stronger rationale for the hypotheses tested and the approach used in the introduction.

In terms of sampling design, it was useful to include the raw datasets as supplementary material but these comprise several large and complex spreadsheets, which don't help the reader understand the sampling. The results section starts with a high-level summary of the sampling [160 trajectories from 58 species or morphospecies, comprised of 93 mimic 197 trajectories, 36 ant trajectories and 31 non-mimic trajectories; body outlines of 304 specimens: 210 mimics, 42 ants and 52 non-mimics; trajectories and morphometric data for 15 species of ant-mimics]. However, as highlighted in the previous review, the finer-grained sampling is important because it can produce biases in the bootstrapping. What is needed is a brief summary describing the sampling in the Methods. For example, for the trajectories, how many of the 58 species were mimics, ants and non-mimics (number of species in each category)? What was the mean, SD and range of trajectories per species in each of these categories? Please provide this detail (samples per species and species per category) for both trajectories and morphology, and the samples per species for the 15 species with both types of data. This information just requires a line or two so would not add much to total length.

Thank you for your comments. We have added the requested summaries as tables, however, as they are quite voluminous (roughly 1 page), we created tables in the appendices (Table A1 & Table A2) and referenced them from the methods. We hope that this is a satisfactory solution.

I also note that we swapped the numbering on Tables 1 & 2 (i.e. Table 1 became Table 2 and vice versa), because the original Table 2 was referenced earlier in the document than the original Table 1.

Apart from adding this information, the reviewers have provided a number of suggestions to improve clarity. These suggestions are minor and straight forward to address.

I think this paper presents significant empirical and conceptual advances and will be an important paper in the field. I look forward to seeing it published.

Reviewer(s)' Comments to Author:

Referee: 1

Comments to the Author(s).

This manuscript is significantly clearer now. The authors are to be commended for dramatically improving the readability of this interesting piece of work. The results and discussion (having sections), in particular, are dramatically improved. Well done!

Thank you for your constructive comments; we agree that the manuscript has been improved as a result.

On that note, I have only very few minor comments, below:

- Note placement of ref 5 on L 47 and 6 on L 54

Erroneous references have been removed.

- L 107, add commas after 'Queensland'

Comma has been added.

- L 197/200. You refer to Fig 1 supp material – this does not exist. I think you are referring simply to Fig. 1

This was very poorly worded and intended to indicate that the trajectories are available (as videos and CSV files) in the supplementary material. We have removed “supplementary material” from the reference and updated the data accessibility section.

- Discussion (no line numbers): “Our learning simulation shows how a naïve predator following one simple rule—avoid anything that seems to be an ant based on what is already known—unavoidably operate” add “s” to “operate”

“s” added.

Referee: 2

Comments to the Author(s).

I found this revised version of the manuscript much improved and appreciate the authors' efforts. The additions and corrections that were made have addressed my main concerns—in particular, the addition of Figure 1, which gives the reader a better sense of the locomotor and morphological features being discussed, and the clarification of which species were used in the analysis. I do have a few corrections, suggestions, and comments regarding this new draft, but I believe that the manuscript (in a lightly revised form) would be worthy of publication in Proceedings B.

Thank you for the helpful comments, which have improved the quality of the paper.

Specific comments/suggestions:

- Line 30 (abstract): The final portion of the abstract states: “while interactions between components of mimicry may limit mimetic accuracy in some mimicry systems.” Perhaps I have simply missed it (or am misinterpreting), but it is not clear to me what this statement is based on—specifically regarding the use of the term “limit”. The section subtitled “Multicomponent hypothesis” in the Discussion seems to clearly state that there is no correlation between traits—and that the current study does not find support for the limitation-related hypothesis that there exists some constraint(s) between ant-like movements and ant-like morphologies. Could another phrase be used in the abstract to avoid this confusion?

Your interpretation is correct, so the phrase has been reworded to avoid this confusion.

- Line 47: It appears that the citation “[5]” shows up here by mistake. Looking at the document showing the changes that were implemented, it seems like this citation might have just accidentally escaped deletion when removing an older piece of text.

The reference has been removed.

- Line 54: Likely the same problem as on Line 47, this time regarding “[6]” at the beginning of the sentence.

The reference has been removed.

- Line 84-85: The behavioral and morphological traits are discussed here as though they are actually singular traits. It might help if it is made clear that they are sets of traits, and for each set a single score was calculated.

We have added a sentence to explain this.

- Table 2: The separations between types of hypotheses in the text is excellent (information limitation, multicomponent hypotheses, and motion-limited hypotheses), but it would be great if that organization could also be present in this table. Without this separation, the table is a bit confusing, particularly because it appears that multiple hypotheses are mutually exclusive but are also supported by the results of the work—a contradiction that disappears when it is clear that these hypotheses address different levels of analysis.

Table 2 now contains visual separators between the types of hypotheses, and has also been re-ordered to the meaning of the separators clearer.

- Discussion: “Motion limited discrimination” section title: Some places in the text the phrase “motion limited” is without a hyphen, while in other cases it is hyphenated (as in the 3rd line of the “Motion limited discrimination” section and in Table 2).

All occurrences are now hyphenated.

- Discussion, “Multicomponent hypotheses”, 4th-6th lines: I like the point being made here, but I think it might help to be a bit more explicit about where the relationship between locomotor power, muscles, and morphology. Something like: “Spider locomotion depends on muscles contained within the legs, but also hydraulic power developed by muscles in the head, and thus hydraulic power depends on head morphology.”

The sentence has been updated as suggested.

- Discussion, “Motion limited discrimination”, 4th line: I would recommend changing “consistency of motion” to “fraction of time spent moving,” or something similar. To me, consistency of motion suggests the pattern and/or frequency of stops, as opposed to the overall duration of movement.

The expression has been changed to “proportion of time moving”, as that is consistent with the wording in the results.

The manuscript with tracked changes follows: